# kNN-CLIP: Retrieval Enables Training-Free Segmentation on Continually Expanding Large Vocabularies

**Zhongrui Gui**[1]    **Shuyang Sun**[1]    **Runjia Li**[1]    **Jianhao Yuan**[1]    **Zhaochong An**[2]
**Karsten Roth**[3]    **Ameya Prabhu**[1,3*]    **Philip Torr**[1*]
[1]**University of Oxford**    [2]**University of Copenhagen**    [3]**University of Tübingen**

Reviewed on OpenReview: `https://openreview.net/forum?id=ZSqP1RT8jC`

## Abstract

Continual segmentation has not yet tackled the challenge of improving open-vocabulary segmentation models with training data for accurate segmentation across large, continually expanding vocabularies. We discover that traditional continual training results in severe catastrophic forgetting, failing to outperform a zero-shot segmentation baseline. We introduce a novel training-free strategy, kNN-CLIP, which augments the model with a database of instance embeddings for semantic and panoptic segmentation that achieves zero forgetting. We demonstrate that kNN-CLIP can adapt to continually growing vocabularies without the need for retraining or large memory costs. kNN-CLIP enables open-vocabulary segmentation methods to expand their vocabularies on any domain with a single pass through the data, while only storing compact embeddings. This approach minimizes both compute and memory costs. kNN-CLIP achieves state-of-the-art performance across large-vocabulary semantic and panoptic segmentation datasets. We hope kNN-CLIP represents a significant step forward in enabling more efficient and adaptable continual segmentation, paving the way for advances in real-world large-vocabulary continual segmentation methods.

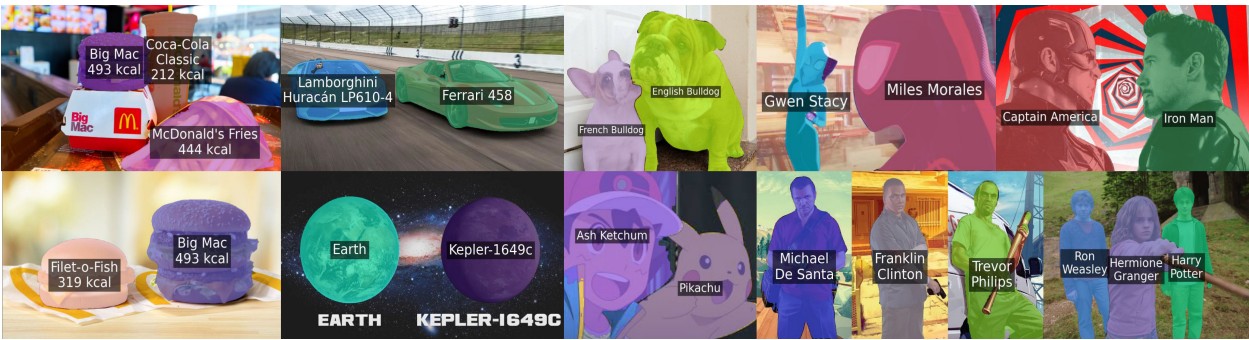

Figure 1: We propose kNN-CLIP to continually expand the vocabulary space of segmentation models. Our approach adapts to concept customization and identifying long-tailed concepts, a known challenge for CLIP models (Udandarao et al., 2024). For concept customization, we build the supporting database for each long-tailed concept efficiently using C2C (Prabhu et al., 2023c). We use EntitySeg (Qi et al., 2022) to generate class-agnostic masks for entities in the database and kNN-CLIP to label these masks. At inference time, we filter the masks from EntitySeg based on confidence thresholds for both the mask and class predictions.

---

*Equal advising; correspondence to Shuyang Sun (`kevinsun@robots.ox.ac.uk`) and Ameya Prabhu (`ameya@prabhu.be`)

# 1 Introduction

The ultimate goal in image segmentation is to accurately segment any image according to a text query across open-ended classes. This task was made more achievable through the rapid advances of vision-language models (VLMs) like CLIP (Radford et al., 2021b), which when applied to semantic and panoptic segmentation, shows promise in handling a broad vocabulary of visual data (Zhou et al., 2021; Rao et al., 2022). However, recent state-of-the-art open-vocabulary segmentation approaches (Yu et al., 2023; Liu et al., 2023c) still rely on fine-tuning with datasets containing fine-grained annotations like masks and boxes. Although they excel in common categories on segmentation benchmarks, they fall short of achieving broad-vocabulary segmentation (Li et al., 2023a; Sun et al., 2023b).

We identify that the success of these models on standard datasets is due to fine-tuning on specific datasets with detailed labels, such as COCO-Stuff (Caesar et al., 2018), which often overlap with the classes of many other benchmarks with fine-grained annotations. For example, COCO-Stuff and ADE-20K share 73 out of 150 classes. Crucially, removing this overlapping vocabulary results in poor segmentation performance (Sun et al., 2023a). Hence, the effective vocabulary of these open-vocabulary models is limited to COCO-Stuff, *i.e.* fine-tuning significantly narrows down the vocabulary size of these models, restricting their ability to generalize to an open-vocabulary of objects and concepts[1]. In this work, we answer the question: *How do we cost-effectively enhance the performance of these models over a continually growing vocabulary space?*

First, we investigate continual training of these open-vocabulary models on data with new categories. We consistently find that continual training of these models causes catastrophic forgetting by overwriting past knowledge from updating a limited, poorly understood parameter space. Hence, we investigate how to expand the knowledge of the segmentation model given in-domain data *without any training*.

Our novel method, kNN-CLIP, simply augments the CLIP segmentation models with a retrieval database that matches images with text descriptions. This database can be updated with new data in a single pass without storing any previous images or requiring training, similar to ACM (Prabhu et al., 2023a). Unlike conventional continual learning methods, kNN-CLIP ensures the model retains knowledge of previously encountered data (zero forgetting), as the distance of a data point to itself in the retrieval database is zero. kNN-CLIP requires a single pass (online), is memory-efficient due to only storing features, and expands its vocabulary with minimal computational resources due to no additional training. The dynamic nature of kNN-CLIP is particularly advantageous for segmentation settings where large volumes of new data are collected daily (Gui et al., 2024), as retraining large foundation models frequently is too expensive.

We demonstrate the effectiveness of kNN-CLIP on state-of-the-art open-vocabulary semantic and panoptic segmentation benchmarks with the FC-CLIP (Yu et al., 2023) model. kNN-CLIP achieves notable performance increases (mIoU) across various challenging datasets: A-847, PC-459, and A-150. Specifically, we see improvements of **+2.6**, **+1.7** and **+7.2** points respectively, demonstrating effective segmentation across a continually-growing vocabulary space. This improvement comes with minimal computational and memory overheads and no training cost. As shown in Fig 1, we also qualitatively demonstrate the effectiveness of our method on tasks beyond current benchmarks, such as concept customization and segmenting long-tailed concepts like celebrities, demonstrating the effectiveness of retrieval for tasks considered challenging for CLIP models (Udandarao et al., 2024). Overall, the key contributions of our study include:

- ***Dynamically Updating Vocabulary Space***: We observe that continual segmentation methods typically underperform compared to zero-shot open vocabulary models. Therefore, we propose adapting the task of continually integrating new classes in continual learning to open vocabulary segmentation models, offering a more feasible solution for continual segmentation.

- ***Training-Free Continual Vocabulary Expansion***: We introduce a novel technique, kNN-CLIP, that continually expands the vocabulary of image segmentation models *beyond the vocabulary space of CLIP* without additional training. This method utilizes a continually expanding support set, recasting the problem of concept prediction as an image retrieval task.

---

[1]This phenomenon of degradation of open-vocabulary performance after fine-tuning CLIP models, known as concept forgetting (Mukhoti et al., 2023), was studied in classification tasks.

- ***Consistent Improvements Across Diverse Datasets***: We present extensive experimental evidence of our approach's effectiveness, demonstrating substantial improvements in semantic and panoptic segmentation across datasets with long-tailed distributions (A-847, PC-459, A-150).

## 2 Related Work

**Retrieval-Augmented Models.** In the NLP domain, Retrieval-Augmented Generation (RAG) has been shown as a promising technique for enhancing Large Language Models (LLM) by leveraging external structured data (Guu et al., 2020; Wu et al., 2022; Borgeaud et al., 2022; Liu et al., 2020; Peters et al., 2019; Yu et al., 2021; Izacard et al., 2022; Ram et al., 2023; Yuan et al., 2024). The dynamic nature of RAG facilitates continuous knowledge updates, enabling models to integrate domain-specific information seamlessly. This is particularly beneficial for knowledge-intensive tasks (Lewis et al., 2020; Wang et al., 2022; Khandelwal et al., 2019; Shi et al., 2023; Petroni et al., 2023). Inspired by its success in NLP, researchers are now exploring the application of RAG in computer vision tasks (Chen et al., 2022a; Marino et al., 2021; Roth et al., 2022; Shen et al., 2022; Sheynin et al., 2022; Yasunaga et al., 2022; Blattmann et al., 2022; Chen et al., 2022b; Liu et al., 2023b; Udandarao et al., 2023; Iscen et al., 2023; Prabhu et al., 2023c; Balazevic et al., 2023). For instance, REACT (Liu et al., 2023b) presents a methodology aimed at retrieving relevant knowledge and learning customized visual modules accordingly for specific domains. Meanwhile, SuS-X (Udandarao et al., 2023) and C2C (Prabhu et al., 2023c) introduce frameworks that utilize retrieved support sets for far more accurate CLIP classification to open-set domains. Although these two works show promise in applying retrieval-based methods to visual perception, they did not discuss how to apply such techniques successfully in visual segmentation. Hummingbird (Balazevic et al., 2023) suggests employing a straightforward non-parametric nearest neighbour retrieval as the decoder for dense scene understanding tasks. Moreover, beyond integrating new visual features, RECO (Iscen et al., 2023) demonstrates the efficacy of incorporating text representations by cross-modal fusion alongside original and retrieved embeddings. However, both Hummingbird (Balazevic et al., 2023) and RECO (Iscen et al., 2023) require training to better integrate retrieval augmentation into their pipelines; in contrast, our method demands no training efforts while providing faster adaptability and strong performance.

**Open Vocabulary Learning in Image Understanding.** Benefiting from advancements in vision-language models (Radford et al., 2021a; Jia et al., 2021), visual models have shown the potential to overcome the constraints of pre-defined closed-set concepts, leading to more flexible open vocabulary image understanding (Li et al., 2023b). This capability is particularly significant for dense prediction tasks (Yu et al., 2023; Rao et al., 2022; Ding et al., 2022; Chen et al., 2023; Karazija et al., 2023; Liang et al., 2023; Li et al., 2022; Xu et al., 2023a;b), where approaches like SimBaseline (Xu et al., 2022b) and OVSeg (Liang et al., 2023) leverage cross-modality supervision from CLIP to align class-agnostic mask proposals with language concepts. Building on these ideas, ODISE (Xu et al., 2023a) and FC-CLIP (Yu et al., 2023) further enhance performance in both semantic and panoptic segmentation by improving visual encoders and mask proposal strategies. Despite these advancements, a critical limitation is the issue of catastrophic forgetting during fine-tuning on concept-restricted dense prediction datasets, which leads to a significant reduction in model vocabulary size (Xu et al., 2022a; Sun et al., 2023a). To mitigate this challenge, training-free methods such as ReCO (Shin et al., 2022) and CaR (Sun et al., 2023a) have been proposed to enable efficient adaptation of CLIP models through recurrent pruning of the label space. While these methods avoid catastrophic forgetting, they still struggle to accommodate continually expanding vocabularies (Wang et al., 2023b; Lüddecke & Ecker, 2022; Xie et al., 2023; Sun et al., 2023b; Seifi et al., 2024). Therefore, in this work, we explore the use of retrieval augmentation to enhance continual open vocabulary image understanding.

**Continual Learning for Semantic Segmentation.** Unlike open vocabulary semantic segmentation, continual learning does not aim to initially encompass an extremely large vocabulary space but instead focuses on preserving the ability to expand this space continuously. However, continual learning is challenged by issues such as catastrophic forgetting and semantic drift. To address these problems, iCaRL (Rebuffi et al., 2017) suggests replaying the most representative samples during the continual learning stage, and subsequent works have sought to optimize the associated memory burden (Cha et al., 2021; Zhu et al., 2023; Wang et al., 2023a). The use of web images for replay in a similar manner has also been explored (Liu et al., 2023a). Additionally, ALIFE (Oh et al., 2022) introduces a feature-replay scheme to reduce memory requirements.

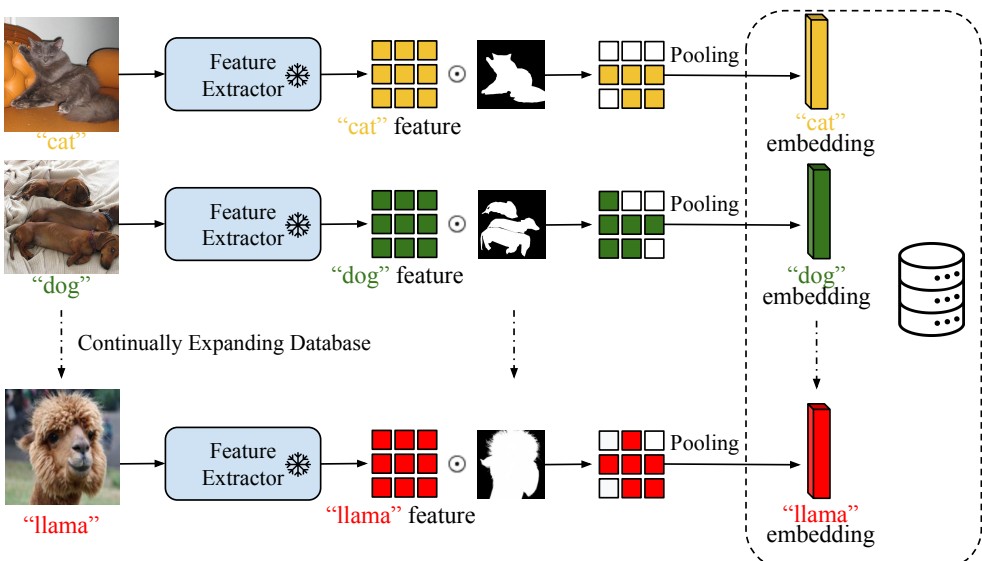

Figure 2: **Dynamic Database Construction.** A key benefit of our methodology is that it allows for the seamless integration of embeddings for new classes into our database, continuously expanding its vocabulary space.

Other research has delved into leveraging memorized features to preserve known knowledge (Yoon et al., 2022; Yu et al., 2020; Wang et al., 2021; Michieli & Zanuttigh, 2021; Douillard et al., 2021). While the use of memorized features shows promise, a significant limitation is that the representation capacity of these features might be insufficient, which can hinder the performance of continual learning (Yuan & Zhao, 2023). Approaches such as employing hyper-class knowledge (Shi et al., 2022) or dynamically updating stored features (Liu et al., 2022; Lin et al., 2022) attempt to mitigate this issue but do not fully circumvent it. Our method, which aims to expand the vocabulary space dynamically, resembles feature-replay methods as we also construct a support set of features. However, our approach differs by operating without the need for additional training, utilizing a support set that stores powerful features learned through self-supervised learning techniques (Caron et al., 2021), with minimal memory requirements.

## 3   kNN-CLIP: Continually Expanding Retrieval-Augmented Dense Prediction

We introduce a novel, training-free framework for continual dense prediction with expanding vocabularies, applicable to diverse domains and across dense prediction tasks. Inspired by the Retrieval-Augmented Generation (RAG) method used in Large Language Models, our framework leverages a customizable embedding database to incorporate domain-specific knowledge directly at inference, with no additional training.

This approach is applicable to a range of dense prediction tasks including semantic and panoptic image segmentation. We aim to augment existing open-vocabulary segmentation models. These models, including FC-CLIP (Yu et al., 2023) which we use as our baseline, usually have a mask proposer trained to generate class-agnostic masks from extracted features. Then, the image CLIP features are pooled using these proposed masks and used to compute similarities with the text CLIP features of class names for classification. Finally, mask predictions and their corresponding class predictions are combined to give the result segmentation masks. We first evaluate the confidence level of CLIP classification results for each query mask. For queries falling below a given confidence threshold, we extract DINO features and use mask pooling to generate query embeddings. These embeddings are then matched against a vectorized database using a kNN search algorithm based on cosine similarities. For the most similar embeddings found, we create a set of confidence-aware pseudo-logits by stacking similarities across class vectors, guided by the labels of these embeddings. The results from this retrieval process are then combined with the initial CLIP results through a given weighting parameter $\lambda$. We now provide details of our training-free method. We first describe the embedding construction process and detail the query mechanisms.

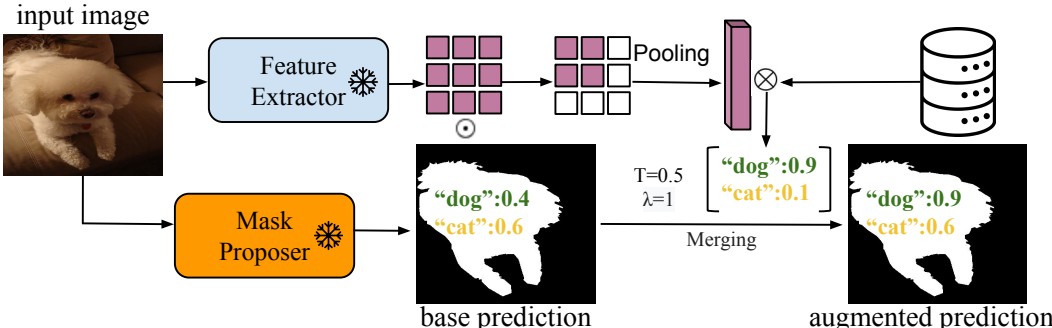

Figure 3: **Retrieval Augmentation from the Database.** By retrieving similar features from the database, we integrate the retrieved information with our previous prediction.

## 3.1 Designing Continually Expanding Embedding Databases

**Database Construction.** To obtain distinctive vectorized representations for the database images, we employ mask-pooling on features extracted by pretrained encoders from the images as shown in Fig 2. Our input consists of an incoming image $(x_i \in \mathcal{R}^{3 \times h \times w})$ with its corresponding $k$ class-agnostic masks, which could be semantic, instance dense masks, or bounding box masks $(M_i \in \mathcal{R}^{k \times h \times w})$ and class annotations $(C_i)$ containing the class $\{c_{ij}\}_{j=1}^k$ corresponding to mask $\{m_{ij}\}_{j=1}^k = M_i$. We extract features of this image using a Vision Transformer (ViT, Dosovitskiy et al. (2021))-based encoder $(\mathcal{E})$, which is pretrained DINOv2 (Oquab et al., 2023) in this case, yielding $h_i = \mathcal{E}(x_i)$. Here, $h_i \in \mathcal{R}^{d \times h' \times w'}$ with $d$ representing the embedding dimension of the feature, and $h'$, $w'$ represent the dimensions of the feature map. The embedding dimensions $h'$, $w'$ are determined by image size divided by the patch size of the vision transformer, thus divided by 14 (the patch size of ViT-G) each, and we reduce the space requirements by storing small feature maps rather than input images. The ground-truth masks $M_i$ are resized to $M_i' \in \mathcal{R}^{k \times h' \times w'}$ to align with the feature map dimensions. By applying mask average pooling along $h'$ and $w'$, the final embedding sets $\{e_{ij}\}_{j=1}^k$ on image $x_i$ is obtained by:

$$E_i = \{e_{ij} = \frac{\sum_{h'} \sum_{w'} (h_i \cdot m_{ij}')}{\sum_{h'} \sum_{w'} m_{ij}'}\}_{j=1}^k, where\ E_i \in \mathcal{R}^{k \times d} \tag{1}$$

Then, the vectorized embedding database is constructed by concatenating the embedding sets and class sets from all images, so that $D_{vec} = \{E_i, C_i\}_{i=1}^n = \{e_{i'}, c_{i'}\}_{i'=1}^{n'}$, if there are $n'$ embeddings in total. New embeddings can also be continually added to the database repeating the above process. Note that *we do not store any images, enabling low storage costs and allowing deletion of previously seen data.*

One appealing characteristic of our framework is the dynamic extendability of the database and the customization of the database construction based on personalized interests. Depending on the purpose specified by the user-defined applications (fine-grained settings such as smart retail, species classification, etc.), the database could be sourced from various domains through manual curation, online web-based collection, or synthetic data sources.

## 3.2 Inference Using the Continually Expanding Embedding Databases

The embedding database established serves as the retrieval source during inference to enhance the capabilities for various downstream visual models, as shown in Fig 3. Given a test image along with a set of query mask proposals $M_q \in \mathcal{R}^{k \times h \times w}$ and corresponding feature embeddings acquired using DINOv2 (Oquab et al., 2023) through the same method outlined in Sec 3.1, we conduct a k-Nearest neighbour (kNN) search for each query $e_q$ within the embedding database $D_{vec}$ as follows:

$$\text{kNN}_k(e_q) = \{(c_j, s_j(\cdot, \cdot)) \,|\, (e_j, c_j) \in D_{vec}, s_j \in \mathbf{s}_{sorted}[:k]\}_{j=1}^k, s_j(e_q, e_j) = \frac{e_q \cdot e_j}{\|e_q\| \|e_j\|} \tag{2}$$

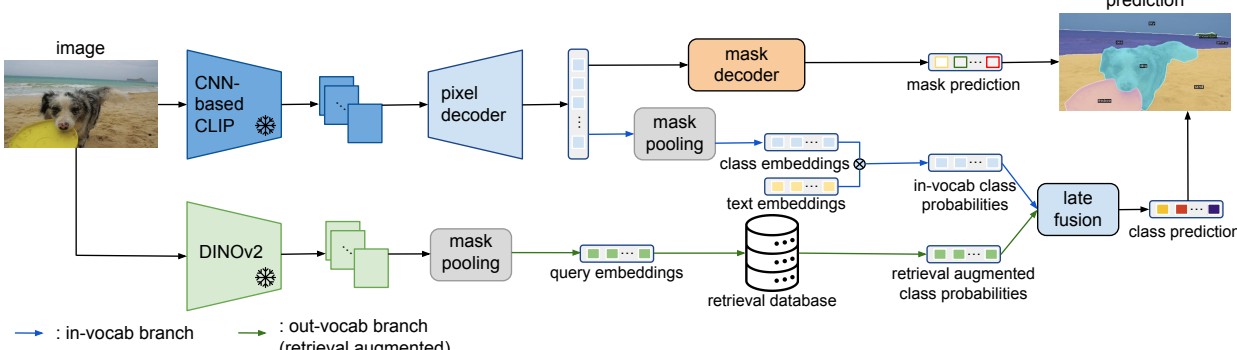

Figure 4: **Augmenting FC-CLIP.** We integrate the retrieval augmentation module to the state-of-the-art segmentation model, FC-CLIP. FC-CLIP includes an in-vocabulary branch and an out-of-vocabulary branch. We don't shown the original out-of-vocabulary branch here for simplicity. We use kNN-CLIP to augment the out-of-vocabulary branch using DINOv2 features and retrieved information.

We then utilize the retrieved sample class labels $\{c_j\}^k$ and cosine similarity scores $\{s_j\}^k$ to construct confidence-aware pseudo-logits $P_{ret} \in \mathcal{R}^C$ for retrieval-based prediction as follows:

$$P_{ret} = \frac{exp\left(\sum_{i=1}^{k}(s_i \cdot \mathbb{I}(c_i = j)) + \epsilon\right)}{\sum_{l=1}^{C} exp\left(\sum_{i=1}^{k}(s_i \cdot \mathbb{I}(c_i = l)) + \epsilon\right)}, \quad \forall j \in [1, C] \tag{3}$$

where we accumulate the class-wise similarity over the class vector and normalize it into a probability distribution through a softmax operation. We consider the higher similarity between instance embeddings with a more frequent retrieval reflects higher confidence in the kNN-based prediction. We then leverage the prediction logits to augment the raw prediction logits $P_{base} \in \mathcal{R}^C$ from CLIP through a class-wise logit modification as follows:

$$P_{final} = \lambda P_{ret} \cdot (1 - \mathbb{I}(max(P_{base}) > T)) + P_{base} \cdot \mathbb{I}(max(P_{base}) > T) \tag{4}$$

where we set a confidence threshold $T$ on raw prediction with the logit lower than it replaced by the corresponding value in retrieval-based prediction.

Our construction of pseudo-logits using a weighting based on cosine similarity only enhances the downstream performance on tasks or classes whose labels are stored in the embedding database. If the true label of a query is not in the database, or if the cosine similarity of a retrieved result is too low, our method will naturally prefer the original predictions. This preserves the broad zero-shot capabilities and the knowledge gained during pretraining, which are adept at making accurate predictions for concepts they were trained on (Udandarao et al., 2024). This flexible design enables us to dynamically expand to larger vocabularies and effectively addresses the recently highlighted issue of concept forgetting (Mukhoti et al., 2023).

## 3.3 Augmenting FC-CLIP

We integrate our methodology into the current state-of-the-art open-vocabulary segmentation model, FC-CLIP (Yu et al., 2023), which contains three main components. The model includes a mask generator, an in-vocabulary classifier, and an out-of-vocabulary classifier. All these components are built on top of a shared CLIP backbone. The pixel decoder and mask decoder generate class-agnostic masks, following Mask2Former (Cheng et al., 2022). Then, by mask-pooling the pixel features from the pixel decoder using the class-agnostic masks proposed, the in-vocabulary classifier gives the in-domain class embeddings; by mask-pooling the CLIP features, the out-vocabulary classifier yields its class embeddings. The results from both classifiers are then fused to give the final prediction.

We use our retrieval augmentation module to enhance the performance of the out-of-vocabulary classifier of FC-CLIP, which is the main reason for the open-vocabulary capability of FC-CLIP. The original out-of-vocabulary classifier leverages the cross-modal CLIP model for open-vocabulary classification. As in Fig 4, to enable retrieval augmentation, we additionally pass the input image into a frozen pretrained DINOv2. We then conduct mask-pooling on the DINO features and obtain our query embeddings as in Sec 3.1. We then compute the retrieval augmented class probabilities as described in Sec 3.2. We fuse the retrieval augmented class probabilities with the original out-of-vocabulary class probabilities derived from CLIP linearly as in Eq. 4. Then, the retrieval augmented results of the out-of-vocabulary classifier are combined with the results of the in-vocabulary classifier to give a final class prediction, following FC-CLIP. We then derive the segmentation masks from mask predictions and class predictions following Mask2Former and FC-CLIP.

## 4 Experiments

We present the results of our training-free approach, aimed at improving open-vocabulary dense prediction for large-scale datasets, including semantic and panoptic segmentation, testing continually expanding vocabularies in customized contexts.

### 4.1 Implementation Details

**Database Construction.** Our database is created by extracting features from each dataset's training sets using DINOv2 (Oquab et al., 2023) with a ViT-Giant architecture, featuring 4 register tokens (Darcet et al., 2023). We select "keys" as our feature representation. We resize the images to $518 \times 518$ and acquire an image feature of dimensions $1536 \times 37 \times 37$, as the patch size of chosen ViT is 14. We then conduct mask average pooling to shrink the feature dimension to 1536. Then, we store the shrunken feature with a dimension of 1536 and its corresponding label $c$ into the database using FAISS. On average, a stored feature occupies 6kB of space. Note that we do not store past images alongside.

**k-Nearest neighbours Search.** We employ FAISS for feature retrieval using the cosine similarity metric. Our method uses a brute force approach for identifying the closest embeddings, but we also explore approximate search methods like Hierarchical Navigable Small Worlds (HNSW) to improve efficiency at inference time (Prabhu et al., 2023a). We shall compare HNSW approximation with exact-search and detail the tradeoffs in our ablations. The number of nearest neighbours retrieved is 16 across dense visual tasks, with further details in the ablation section.

### 4.2 Catastrophic Forgetting Restricts Open-Vocabulary Capabilities of Models

**Motivation.** We study how the open-vocabulary performance of dense predictors varies as these models are trained to recognize new classes. Specifically, we compare their segmentation performance before and after training and report performance deterioration.

**Setup.** We adopt a class-incremental continual learning setting adopted from CoMFormer(Cermelli et al., 2023). We benchmark this on the semantic segmentation benchmark of COCO Panoptic as a base dataset (having 330K images and 80 classes) and ADE20K (Zhou et al., 2019) as a continual dataset (having 27K images and 150 classes). We start from the model weights of FC-CLIP, which was trained on COCO Panoptic, and then incrementally update the base model on ADE20K, with 5, 10 and, 30 classes per timestep and a computational budget of one epoch (Prabhu et al., 2023b;a). We follow Mask2Former (Cheng et al., 2022) to use their loss function on an FC-CLIP (Yu et al., 2023) model and evaluate mIoU across all classes in COCO Panoptic and ADE20K respectively after continual training.

**Results.** As shown in Table 1, when using naïve continual learning baseline, regardless of the number of incremental classes, we observe consistent performance degradation of around 6 mIoU and 24 mIoU on the continual learning dataset ADE20K and the base dataset COCO-Panoptic respectively. The degradation on ADE20K can be attributable to catastrophic forgetting when learning across timesteps. Moreover, this causes concept forgetting, i.e., degradation of the open-vocabulary performance by a large margin, as measured by performance on COCO-Panoptic. This indicates that there is a pressing need for techniques to allow

Table 1: **Vocabulary Narrowing FC-CLIP.** We observe that further training FC-CLIP within a computational budget leads to significantly poorer performance on the downstream ADE20K dataset, while our retrieval augmentation method circumvents the problem in a compute efficient manner. We also observe significant vocabulary narrowing in COCO Panoptic dataset, which serves as the original training set of FC-CLIP.

| Method | Base Dataset | Continual Dataset | Incremental Class | COCO Panoptic | ADE20K |
|---|---|---|---|---|---|
| FC-CLIP (Yu et al., 2023) | COCO Panoptic | - | - | 63.7 | 34.1 |
| FC-CLIP (Yu et al., 2023) | COCO Panoptic | ADE20K | 5 | 43.0 (-20.7) | 28.9 (-5.2) |
| | COCO Panoptic | ADE20K | 10 | 38.9 (-24.8) | 27.7 (-6.4) |
| | COCO Panoptic | ADE20K | 30 | 39.6 (-24.1) | 26.3 (-7.8) |
| kNN-FC-CLIP (Ours) | COCO Panoptic | ADE20K | 1 | **65.5 (+1.8)** | **41.3 (+7.2)** |

Table 2: **Comparisons with Continual Learning.** Retrieval augmentation circumvents catastrophic forgetting occuring in continual learning.

| Method | Base Dataset | Continual Dataset | Incremental Class | ADE20K (mIoU) |
|---|---|---|---|---|
| FC-CLIP (Yu et al., 2023) | COCO Panoptic | - | - | 34.1 |
| FC-CLIP (Yu et al., 2023) | COCO Panoptic | ADE20K (150 classes) | 5 | 28.9 (-5.2) |
| | COCO Panoptic | ADE20K (150 classes) | 10 | 27.7 (-6.4) |
| | COCO Panoptic | ADE20K (150 classes) | 30 | 26.3 (-7.8) |
| Mask2Former (Cheng et al., 2022) | ADE20K (100 classes) | ADE20K (50 classes) | 5 | 26.3 |
| | ADE20K (100 classes) | ADE20K (50 classes) | 10 | 31.0 |
| | ADE20K (100 classes) | ADE20K (50 classes) | 50 | 38.1 |
| CoMFormer (Cermelli et al., 2023) | ADE20K (100 classes) | ADE20K (50 classes) | 5 | 30.9 |
| | ADE20K (100 classes) | ADE20K (50 classes) | 10 | 32.3 |
| | ADE20K (100 classes) | ADE20K (50 classes) | 50 | 38.4 |
| kNN-FC-CLIP (Ours) | COCO Panoptic | ADE20K | 1 | **41.3 (+7.2)** |

segmentation models to continually expand their vocabulary across novel concepts without losing their open-vocabulary segmentation capabilities. We show that our proposed training-free method kNN-CLIP allows effective alleviation of concept forgetting leading to significant performance improvement of **+7.2** mIoU in ADE20K and **+1.8** mIoU in COCO Panoptic.

## 4.3 Comparison with Continual Segmentation Approaches

**Setup.** We compare our method under the setting of continual learning against the popular supervised method, Mask2Former (Cheng et al., 2022) and the current state-of-the-art continual learning method, CoMFormer (Cermelli et al., 2023). We use the same experimental protocol as previously described for our method. These supervised models were initially trained on the first 100 classes and subsequently updated incrementally with the next 50 classes. We then evaluated the performance of these models using the mean Intersection over Union (mIoU) metric across all 150 classes from the ADE20K dataset.

**Results.** As shown in Table 2, Mask2Former (Cheng et al., 2022) and CoMFormer (Cermelli et al., 2023) exhibit large performance degradation when we reduce the number of incremental classes within 50 classes. In contrast, kNN-CLIP demonstrates a substantial advantage in preserving previous knowledge and mitigating knowledge loss, and it is inherently unaffected by the number of incremental classes. This resilience is attributed to our dynamic and continuously expanding embedding database, which significantly enhances our method's effectiveness in continual learning scenarios.

## 4.4 Retrieval Enhances Panoptic Segmentation

**Setup.** Our exploration extends to panoptic segmentation, further validating the discernability of instance-level open-vocabulary recognition. We evaluate our method on ADE20K and COCO Panoptic datasets. To

Table 3: **Open-vocabulary panoptic segmentation performance.** Retrieval augmentation showcases improvements in panoptic segmentation.

| Method | ADE20K | | | COCO Panoptic | | |
|---|---|---|---|---|---|---|
| | PQ | AP | mIoU | PQ | AP | mIoU |
| ODISE (Xu et al., 2023a) | 22.6 | 14.4 | 29.9 | 55.4 | **46.0** | 65.2 |
| ODISE(caption) | 23.4 | 13.9 | 28.7 | 45.6 | 38.4 | 52.4 |
| FC-CLIP(baseline) | 26.8 | 16.8 | 34.1 | 54.4 | 44.6 | 63.7 |
| kNN-FC-CLIP(Ours) | **29.6** | **17.5** | **41.3** | 54.8 | 44.8 | **65.5** |
| Δ *w.r.t baseline* | +2.8 | +0.7 | +7.2 | +0.4 | +0.2 | +1.8 |

Table 4: **Open-vocabulary semantic segmentation performance.** Retrieval augmentation demonstrates state-of-the-art performances on open-vocabulary semantic segmentation.[3]

| Method | Training Dataset | A-847 | PC-459 | A-150 | PC-59 | PC-21 |
|---|---|---|---|---|---|---|
| SimBaseline (Hsia et al., 2022) | COCO Stuff | - | - | 15.3 | - | 74.5 |
| ZegFormer (Ding et al., 2022) | COCO Stuff | - | - | 16.4 | - | 73.3 |
| LSeg+ (Li et al., 2022) | COCO Stuff | 3.8 | 7.8 | 18.0 | 46.5 | - |
| OVSeg (Liang et al., 2023) | COCO Stuff | 9.0 | 12.4 | 29.6 | 55.7 | - |
| SAN (Xu et al., 2023b) | COCO Stuff | 13.7 | 17.1 | 33.3 | 60.2 | - |
| OpenSeg (Ghiasi et al., 2022) | COCO Panoptic + COCO Caption | 6.3 | 9.0 | 21.1 | 42.1 | - |
| ODISE (caption) (Xu et al., 2023a) | COCO Panoptic + COCO Caption | 11.0 | 13.8 | 28.7 | 55.3 | 82.7 |
| MaskCLIP (Zhou et al., 2021) | COCO Panoptic | 8.2 | 10.0 | 23.7 | 45.9 | - |
| ODISE (Xu et al., 2023a) | COCO Panoptic | 11.1 | 14.5 | 29.9 | 57.3 | 84.6 |
| FC-CLIP(baseline) (Yu et al., 2023) | COCO Panoptic | 14.8 | 18.2 | 34.1 | 58.4 | 81.8 |
| kNN-FC-CLIP(ours) | COCO Panoptic | **17.4** | **19.9** | **41.3** | **62.8** | **85.3** |
| Δ *w.r.t baseline* | | +2.6 | +1.7 | +7.2 | +4.4 | +3.5 |

assess the performance comprehensively, we employ three critical metrics: Panoptic Quality (PQ), Average Precision (AP), and mean Intersection over Union (mIoU).

**Results.** As detailed in Table 3, retrieving information from additional support sets dramatically increases performance on ADE20K, with PQ, AP, and mIoU rising by **+2.8**, **+0.7**, **+7.2** respectively without any training. Overall, the performance of open-vocabulary panoptic segmentation can be significantly boosted without any training.

**Improving Base Dataset Performance.** We highlight that we observed improvements of **+0.4** in PQ, **+0.2** in AP, and **+1.8** in mIoU for COCO Panoptic dataset itself. This highlights that using retrieval even from the baseline model's training dataset, in this case, COCO Panoptic, significantly enhances segmentation accuracy. Our method complements advances in open-vocabulary panoptic segmentation tasks, with these results demonstrating the consistent improvements achieved.

### 4.5 Retrieval Enhances Semantic Segmentation

**Setup.** Our study extends studying the impact of training-free continual vocabulary expansion of kNN-CLIP to semantic segmentation, testing its efficacy across dense prediction tasks. We analyze the performance of kNN-CLIP over five diverse datasets: ADE20K and A847 (Zhou et al., 2019), containing 27K images, Pascal Context(PC)-59/459 (Mottaghi et al., 2014), and Pascal VOC-21 (Everingham et al., 2010) containing 10K images. These datasets include pixel-level annotation on various granularity covering a wide-range of semantic concepts from 150, 847, 21, 59, and 459 classes, respectively. We use the same FC-CLIP backbone as in Sec 4.1. For all benchmarks, we leverage the mIoU metric to evaluate segmentation performance.

---

[3]Our method aims to continually enhance the performance of open-vocabulary segmentation models. For evaluation, we use the training set of each dataset, which is unseen by other methods, to construct our embedding database. Our approach significantly improves the performance of zero-shot baselines in a continual learning setting.

Table 5: **Comparisons with Hummingbird.** Our method adopts a similar nearest neighbors retrieval module as Hummingbird but shows significant improvements.

| Dataset | Hummingbird | Hummingbird++ | kNN-CLIP (Ours) |
|---|---|---|---|
| PASCAL VOC | 76.9 | 77.3 | 85.3 (+8.0) |
| ADE20K | 35.0 | 35.8 | 41.3 (+5.5) |

Table 6: **(Left) Confidence Threshold and Confidence Weighting Ablations.** Comparisons of different choices of hyper-parameters on A-847. **(Right) Number of Retrieved Nearest Neighbors Ablations.** Comparisons of different choices of number of retrieved nearest neighbors on A-847 with $T = 0.7$ and $\lambda = 1.2$.

| No. Neighbors (k) | Confidence Weighting | Confidence Threshold | ADE-847 (mIoU) |
|---|---|---|---|
| - | - | - | 14.79 |
| 16 | 1.0 | 0.65 | 17.28 |
| 16 | 1.2 | 0.65 | 17.27 |
| 16 | 1.4 | 0.65 | 17.33 |
| 16 | 1.0 | 0.7 | 17.43 |
| **16** | **1.2** | **0.7** | **17.45** |
| 16 | 1.4 | 0.7 | 17.42 |
| 16 | 1.0 | 0.75 | 17.35 |
| 16 | 1.2 | 0.75 | 17.42 |
| 16 | 1.4 | 0.75 | 17.41 |
| 16 | 1.0 | 0.85 | 17.25 |

| Number of Neighbors | mIoU |
|---|---|
| - | 14.79 (baseline result) |
| 1 | 13.77 |
| 4 | 14.84 |
| 8 | 16.60 |
| **16** | **17.45** |
| 32 | 17.24 |
| 64 | 16.64 |

**Results.** The effectiveness of our method is clearly demonstrated in Table 4, showcasing significant improvements across various benchmarks. When compared to the baseline FC-CLIP model, we boost mIoU by **+2.6**, **+1.7**, **+7.2**, **+4.4**, **+3.5**, across A-847, PC-459, A-150, PC-59, and PC-21, respectively. With retrieval augmentation, the performance on long-tailed datasets, such as A-847 and PC-459, is generally improved. Our method aims to complement advances in open-vocabulary semantic segmentation and these results underscore the robustness and adaptability of our approach in handling complex segmentation tasks.

## 4.6 Comparisons with Retrieval-based Approaches

**Comparison.** We also compare our results with Hummingbird (Balazevic et al., 2023), which proposes a novel pretraining framework that is suitable for fast downstream adaptions. Hummingbird employs nearest neighbour retrieval for predicting the classes of patches in target images, while we use kNN to augment the prediction of each proposed class agnostic mask. We compare our semantic segmentation results with Hummingbird's best-performing model on Pascal VOC and ADE20K. The results of both methods are derived by performing nearest neighbour retrieval from the same evaluated downstream datasets without any further training.

**Results.** As shown in Table 5, our method outperforms the best Hummingbird model with an increase of **+8.0** and **+5.5** mIoU in PASCAL VOC and ADE20K respectively. The result demonstrates that predictions based on class agnostic masks are more robust, and the information in image patches might not be sufficient for nearest neighbour retrieval. The result further showcases the strong performance and fast adaptability of our straightforward method over other retrieval-based approaches.

## 4.7 Ablations

**Effect of Confidence Threshold and Confidence Weightings.** We examine the impact of varying hyper-parameter settings on performance, focusing on two key parameters: the confidence threshold and the retrieved confidence weightings. The selection of the confidence threshold is primarily influenced by the accuracy of the kNN retrieval module across different datasets. Our experiments are carried out on the A-847

Table 7: **Ablations on Approximate Nearest Neighbors Search.** Comparisons of Inference FPS using Brute Force KNN and HNSW built with efConstruction = 100 and efSearch = 64 for COCO and ADE20K datasets. All results were obtained using a single A40 GPU with CUDA 11.8 and PyTorch 2.0.0. The average runtime was calculated across the entire validation set and includes post-processing time.

| method | ADE20K | | COCO | |
|---|---|---|---|---|
| | FPS | mIoU | FPS | mIoU |
| ODISE | 0.41 | 29.9 | 0.39 | 65.2 |
| FC-CLIP | 2.86 | 34.1 | 2.84 | 63.7 |
| RA-FC-CLIP (Brute Force) | 1.31 | 41.3 (+7.2) | 0.83 | 65.5 (+1.8) |
| RA-FC-CLIP(HNSW) | 1.52 (×1.16) | 34.2 (+0.1) | 1.51 (×1.82) | 64.5 (+0.8) |

dataset. As shown in Table 6 (Left), the optimal performance, measured in terms of the highest mIoU, is achieved when the confidence threshold, denoted as $T$ in Sec 3.2, is set to 0.7, and the confidence weighting, represented by $\lambda$ in the same section, is set at 1.2.

**Effect of Numbers of neighbours Retrieved.** In dense visual tasks, such as semantic segmentation, our approach is to enhance the pseudo-logits for classification of each query mask by incorporating information from the 16 closest neighbours. Additionally, we conduct experiments to examine the impact of varying the number of nearest neighbours retrieved. The results are shown in Table 6 (Right). These experiments are carried out using the previously determined optimal hyper-parameters, where $T$ is set to 0.7 and $\lambda$ is set to 1.2.

**Approximate Nearest Neighbour Search.** To optimize our inference time, we explore approximate nearest neighbour search algorithms to reduce runtime. We apply Hierarchical Navigable Small World to our feature retrieval module. As expected, we observe a trade-off between inference time and model performance. As in Table 7, it can be seen that adopting HNSW leads to a faster runtime but makes compromises with mIoU. With the medium size of the support set, such as ADE20K including approximately 220k features, brute force search is preferred; if the size of the support set is large, as for COCO, which has 860k features, approximate nearest neighbour search offers a better balance between inference time and accuracy.

**Inference Time.** We assess the inference time based on our implementation of FC-CLIP, taking into account only the time required to infer an image. The retrieval database is preconstructed, ensuring that our evaluation focuses solely on the inference cost. By integrating an additional feature extractor, DINOv2, into the existing architecture, we observe a compromise in inference speed as shown in Table 7. Furthermore, the kNN search module contributes to the slower inference time due to our utilization of brute force search. We observe that inference speed can be enhanced by implementing approximate neighbour search techniques as alternatives to brute force search, but approximate neighbour search leads to a slightly worse performance.

## 5  Conclusion

In our study, we examine the constraints related to the vocabulary used in earlier open-vocabulary methods that rely on CLIP, and we also highlight how performance diminishes when Vision-Language Models (VLMs) are fine-tuned using downstream annotations. Instead of adopting continual learning approaches, we introduce a novel method called kNN-CLIP, which does not require training and utilizes nearest neighbour retrieval to predict dense visual tasks. To accommodate ever-growing vocabularies, we employ a dynamic retrieval database, circumventing the need for retraining. Our approach demonstrates strong performance across several datasets and showcases significant improvements over other methods.

**Broader Impact.** We aim to enable dense prediction models across a large and expanding vocabulary of classes. However, significant advancements are required before deployment in critical applications like self-driving cars or medical imaging, pending thorough testing and validation. Additionally, to prevent misuse, we explicitly prohibit our approach's use in military or surveillance contexts.

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
