# OpenReview forum: "kNN-CLIP: Retrieval Enables Training-Free Segmentation on Continually Expanding Large Vocabularies"
_TMLR — Accepted by TMLR_

### Review · Reviewer_XLqz · 2024-04-24

**Summary Of Contributions:**

The paper introduces kNN-CLIP, a method for Open-Vocabulary Segmentation in a continual training setting (continually-growing vocabulary). Its main contribution is proposing a method to prevent catastrophic concept forgetting.  This method (kNN-CLIP) adds a retrieval mechanism to CLIP such that novel concepts (extended vocabulary) are not stored in CLIP's weights but in the concept bank over which retrieval is performed. This approach is related to retrieval-augmented generation in large-language models. The author study the proposed kNN-CLIP on a number of segmentation benchmarks and compare it to baselines, to which it performs favourably.

**Audience:**

Yes

**Broader Impact Concerns:**

No concerns on broader impact.

**Claims And Evidence:**

Yes

**Requested Changes:**

- Page 2 third paragraph: "we can guarantee that the model never forgets previously seen data": please specify what specifically your work guarantees (in a more formal way)
- Figure 2: using the term "support vector" here is suboptimal, since it is never introduced in the main document and also is too reminiscent on SVMs
 - The introduction of Section 3 mentions the usage of CLIP+DINO, but the rest of the section never clarifies where and how they are used.
 - "The embedding dimensions h′, w′ are determined by image size divided by the patch size of the vision transformer, thus usually shrunken by a magnitude of 10 each,": isn't this usually 8, 16 or 32?
 - "The ground-truth masks Mi are resized to M′ i ∈ Rk×h′×w′ to align with the feature map dimensions.": how are they resized? NN-interpolation? wouldn't a soft interpolation (bilinear or similar) make more sense here?
 - The notation in Equation 1 is  not sound: (i) there should not be an equal sign in the definition of the set. (ii) "R^{k×d} = {...}" this is wrong, the defined set is not equal to R^{k×d} but to E_i. I understand what the authors want to say, but I would ask them to fix the notation here.
 - Equation 2: there is nothing in the notation which makes the set "kNN_k(eq)" actually contain the k-nearest neighbours. There is no sorting of the index j according to the respective cosine similarities.
 - Equation 3: please clarify why no temperature parameter is used. Usually, CLIP uses a temperature when mapping from cosine similarities to confidences
 - Equation 4: please clarify why using a hard-threshold-based combination is preferable over a soft-thresholding
 - In the ablations  (Table 6)  both the number of neighbours and the confidence threshold are varied. However, they are only varied independently and not jointly. E.g could it be that 64 neighbours and a higher threshold would yield a better result than the currently best combination of k=16 and T=0.7?
- "ViT-Giant architecture, featuring 4 register tokens." please cite the work introducing register tokens.
- Please clarify what a "geometric ensemble" is.
- "We first train the models like FC-CLIP [...]": please make the paper self-contained by explaining briefly how FC-CLIP was trained.
- Table 2 (or one of the others): please report the memory overhead of kNN-CLIP for the retrieval database. Even if it is small, it is relevant to know "how small".
 - Number of neighbours in retrieval (Table 7): does it make sense to choose this constantly regardless of the size of the retrieval database? Specifically initially, when the database is very small, retrieving 16 neighbours sounds like a lot.
- since inference time drops substantially with kNN-CLIP: Is using DINO ViT-G really required or would smaller ViTs be an alternative?

**Strengths And Weaknesses:**

Strengths:
 - The work studies an interesting and relevant problem setting (Open-Vocabulary Segmentation with a Growing Vocabulary)
 - The approach makes intuitive sense, is simple, and a logical continuation of prior work
 - Overview over related works is solid
 - The authors evaluate the method extensively and kNN-CLIP performs favourably to baselines.

Weaknesses:
 - The exposition of the method (Section 3) lacks clarity (e.g. on the usage of CLIP and DINO)
 - Mathematical notation lacks rigour (see comments on requested changes)

Overall, I think this is a promising work but it needs more care in writing and exposition. See requested changes below.

---

> ### Author Response · Authors · 2024-05-27
>
> Thank you for your thoughtful feedback and constructive suggestions on our manuscript. We are grateful for your recognition of the novel problem setting and the simplicity and logical coherence of our approach. We thank the reviewer for providing detailed feedback to improve clarity. It has improved clarity in areas of the draft! We detail our responses below:
>
> ### **Guarantees regarding "the model never forgets previously seen data"**
> - "Page 2 third paragraph: "we can guarantee that the model never forgets previously seen data": please specify what specifically your work guarantees (in a more formal way)"
>
> We agree and have revised the sentence to improve clarity: “In contrast to conventional continual learning methods, we ensure that the model perpetually retains knowledge of previously encountered data, as the distance of a data point to itself in the retrieval database is zero, ...”.
>
> ### **Replacing "support vector"**
> - "Figure 2: using the term "support vector" here is suboptimal, since it is never introduced in the main document and also is too reminiscent on SVMs"
>
> We agree with the concern about the term "support vector" and its association with SVMs. We have replaced it with "embedding" throughout the manuscript.
>
> ### **Usage of CLIP and DINO in Section 3**
> - "The introduction of Section 3 mentions the usage of CLIP+DINO, but the rest of the section never clarifies where and how they are used."
>
> We agree and have clarified the usage of CLIP and DINO in subsection 3.3. We state it here for easier reference: “The original out-of-vocabulary classifier leverages the cross-modal CLIP model for open-vocabulary classification. To enable retrieval augmentation, as shown in Fig 4, we pass the input image into a frozen pretrained DINOv2. We then perform mask-pooling on the DINOv2 features to obtain our query embeddings and compute the retrieval-augmented class probabilities as described in Sec 3.3. These probabilities are then linearly combined with the out-of-vocabulary class probabilities derived from CLIP, as per Eq. 4.”
>
> ### **Clarification of Embedding Dimensions**
> - ""The embedding dimensions h′, w′ are determined by image size divided by the patch size of the vision transformer, thus usually shrunken by a magnitude of 10 each,": isn't this usually 8, 16 or 32?"
>
> Thank you for the correction. We have revised the sentence for accuracy: “The embedding dimensions $h'$, $w'$ are determined by the image size divided by the patch size of the vision transformer, which is 14 (the patch size of ViT-G).”
>
> ### **Resizing Ground-Truth Masks**
> - ""The ground-truth masks Mi are resized to M′ i ∈ Rk×h′×w′ to align with the feature map dimensions.": how are they resized? NN-interpolation? wouldn't a soft interpolation (bilinear or similar) make more sense here?"
>
> We use NN-interpolation to resize the ground-truth masks. Although we tested soft interpolation methods like bilinear, the difference was minor due to the averaging of features over the masks. Therefore, we opted for the simpler NN-interpolation.
>
> ### **Notation in Equation 1**
> - "The notation in Equation 1 is not sound: (i) there should not be an equal sign in the definition of the set. (ii) "R^{k×d} = {...}" this is wrong, the defined set is not equal to R^{k×d} but to E_i. I understand what the authors want to say, but I would ask them to fix the notation here."
>
> We agree and have corrected the notation in Equation 1 by removing the equal sign in the definition of the set and properly defining it as $E_i$ instead of $R^{k×d}$.
>
> ### **Clarification of Equation 2**
> - "Equation 2: there is nothing in the notation which makes the set "kNN_k(eq)" actually contain the k-nearest neighbours. There is no sorting of the index j according to the respective cosine similarities."
>
> We agree and have fixed the notation in Equation 2 to explicitly show that the set "kNN_k(eq)" contains the k-nearest neighbors, including sorting the index $j$ according to the cosine similarities.
>
> ### **Temperature Parameter in Equation 3**
> - "Equation 3: please clarify why no temperature parameter is used. Usually, CLIP uses a temperature when mapping from cosine similarities to confidences"
>
> We do not use a temperature parameter to maintain consistency with our baseline method, FC-CLIP. However, we agree that incorporating and tuning a temperature parameter could be beneficial.

---

> ### Author Response · Authors · 2024-05-27
>
> ### **Hard-Threshold-Based Combination in Equation 4**
> - "Equation 4: please clarify why using a hard-threshold-based combination is preferable over a soft-thresholding"
>
> We use a hard threshold because the retrieval-augmented probabilities and the class probabilities from CLIP are not on the same scale. For instance, a probability of 0.5 in the retrieval-augmented probabilities does not equate to a probability of 0.5 in the CLIP class probabilities, making direct comparisons challenging. We think the obtained class probabilities should be used within their respective domains or as a metric to compare across classes for a single prediction.
>
> ### **Ablations in Table 6**
> - "In the ablations (Table 6) both the number of neighbours and the confidence threshold are varied. However, they are only varied independently and not jointly. E.g could it be that 64 neighbours and a higher threshold would yield a better result than the currently best combination of k=16 and T=0.7?"
>
> In our experiments, we observed that label retrieval accuracy decreases after 16 neighbors. The number of neighbors is not correlated with the confidence threshold, so we optimized them separately. The optimal number of nearest neighbors depends on the specific datasets and the size of the embedding database.
>
> ### **Citation of ViT-Giant Register Tokens**
> - ""ViT-Giant architecture, featuring 4 register tokens." please cite the work introducing register tokens."
>
> Thank you for pointing this out. We have added the appropriate citation for the introduction of register tokens in the ViT-Giant architecture.
>
> ### **Clarification of "Geometric Ensemble"**
> - "Please clarify what a "geometric ensemble" is."
>
> Sorry about the confusion. We have replaced the term "geometric ensemble" with "late fusion" throughout the manuscript for clarity, as it is a more intuitive term commonly used in the literature.
>
> ### **Training Description of FC-CLIP**
> - ""We first train the models like FC-CLIP [...]": please make the paper self-contained by explaining briefly how FC-CLIP was trained."
>
> We have added a subsection 3.3 explaining how to adapt our method to the FC-CLIP model to make the paper self-contained.
>
> ### **Memory Overhead of kNN-CLIP**
> - "Table 2 (or one of the others): please report the memory overhead of kNN-CLIP for the retrieval database. Even if it is small, it is relevant to know "how small"."
>
> We provide information about the memory overhead here: ADE20K requires 968MB for 165,120 embeddings, A847 requires 1.4GB for 231,666 embeddings, and COCO requires 4.7GB for 816,269 embeddings. We add in the work that the average storage needed per embedding is approximately 6kB for ease-of-calculation across benchmarks.
>
> ### **Number of Neighbors in Retrieval (Table 7)**
> - "Number of neighbours in retrieval (Table 7): does it make sense to choose this constantly regardless of the size of the retrieval database? Specifically initially, when the database is very small, retrieving 16 neighbours sounds like a lot."
>
> We acknowledge that the number of neighbors retrieved is relevant to the size of the database. In our analysis of large embedding databases from datasets such as COCO and ADE, retrieving 16 neighbors was reasonable for a large volume of features.
>
> ### **Inference Time with Smaller ViTs**
> - "since inference time drops substantially with kNN-CLIP: Is using DINO ViT-G really required or would smaller ViTs be an alternative?"
>
> While using smaller ViTs could be an alternative, we highlight the majority of the extra inference time is due to the nearest neighbors search. Therefore, using smaller ViTs might impact performance but would not significantly reduce inference time.

---

> > ### Comment · Reviewer_XLqz · 2024-06-03
> > **Follow-up question**
> >
> > I would like to thank the authors for taking my feedback into account and addressing my main concerns. One remaining concern is not  "incorporating and tuning a temperature parameter" for the retrieval class confidences ($P_{ret}$). This is not relevant for accuracy scores but calibration of class confidences plays a crucial rule for instance when doing late fusion of class scores as in this paper (that is fusing $P_{ret}$ and $P_{base}$ as in Equation 4). This lack of calibration could also explain the authors' own finding that "retrieval-augmented probabilities and the class probabilities from CLIP are not on the same scale" and thus a hard thresholding is required. Can the authors comment on this?

---

> > > ### Author Response · Authors · 2024-06-08
> > > **Our findings**
> > >
> > > Thank you for the insightful comment on incorporating a temperature parameter. Due to the dynamic nature of our database, we found that we would need to constantly adjust the temperature parameter as new embeddings are added. However, that lead to minimal differences. We found our late fusion implementation uses a lambda parameter to balance retrieval-augmented scores with CLIP scores, which corrects for the same effect post-hoc. We believe this method is more effective than temperature scaling while achieving the same result.

---

### Review · Reviewer_2MDe · 2024-05-14

**Summary Of Contributions:**

The manuscript proposes to augment open vocabulary segmentation models with a database of embeddings ($D_{vec}$) for the task of expanding their segmentation capabilities to new categories. This is achieved by combining the segmentation predictions of the base model with a prediction computed from the KNN embeddings retrieved from $D_{vec}$.

The manuscript reports that without this augmentation, the base model, which is normally fine-tuned on a segmentation dataset, performs poorly on categories not labeled in that segmentation dataset. The KNN approach is presented within the framework of class incremental learning.

**Audience:**

Yes

**Broader Impact Concerns:**

Segmentation can be used in sensitive applications such as medical, military, and surveillance. A broader impacts statement would be useful.

**Claims And Evidence:**

No

**Requested Changes:**

Concerns have already been raised in the section above.

**Strengths And Weaknesses:**

## Strengths

- The experiments cover several types of segmentation datasets. A comparison against supervised method method (Mask2Former) and class incremental method (CoMFormer) helps contextualize the methods performance. A comparison is made to Hummingbird is section 4.6 (Table 5).

- The ablations evaluate the most obvious hyper-parameters ($k$, $\lambda$, and $T$).

- The main strength is the demonstration that a simple nearest neighbour approach is able to boost performance on a sophisticated task like open vocabulary panoptic segmentation.

## Weaknesses

- Figure 1 teases concept customization, but this is not discussed nor evaluated in the manuscript. As it has been advertised centrally in page 1 figure 1 it deserves an evaluation.

- The numbers reported for Mask2Former in Table 2 are far below the numbers reported in the Mask2Former paper. Which table in the Mask2Former paper (https://arxiv.org/pdf/2112.01527) should I look at to compare? Which type of segmentation task is being evaluated in section 4.2 / Table 2?

- The details of how Hummingbirg and the proposed method differ have not been discussed making the comparison in Table 5 less useful to the reader. Please elaborate in the manuscript (or appendix).

## Clarification questions

- In section 4.1, does the "linear combination of original predicted class probabilities, with the retrieved prediction class probabilities" refer to Eq. 4?

- In the same paragraph, which equation does the "geometric ensemble" refer to?

- In Table 4, which dataset is used to compute the database of embeddings ($D_{vec}$)? Are all methods in the table seeing the same number and types of annotations?

---

> ### Author Response · Authors · 2024-05-27
>
> Thank you for your insightful feedback and constructive comments on our manuscript. We appreciate your acknowledgment of our comprehensive experimental setup. We seek  to address the concerns, clarify misunderstandings and answer questions below:
>
> ### **Weaknesses**
> ### **Figure 1 and Concept Customization**
> 1. "Figure 1 teases concept customization, but this is not discussed nor evaluated in the manuscript. As it has been advertised centrally in page 1 figure 1 it deserves an evaluation."
>
> We acknowledge the reviewer's concern regarding the lack of concept customization in Figure 1. We respectfully argue that we have conducted extensive empirical evaluations across various datasets, which can be regarded as a customized set of concepts. We point out that it might be challenging to evaluate the customized concepts presented in the teaser, as no such benchmark exists for this type of research. Making a dataset for evaluating of concept customization in the field of open-vocabulary segmentation remains an open problem, currently out-of-scope of our work.
>
> ### **Mask2Former Comparison in Table 2**
> 2. "The numbers reported for Mask2Former in Table 2 are far below the numbers reported in the Mask2Former paper. Which table in the Mask2Former paper (https://arxiv.org/pdf/2112.01527) should I look at to compare? Which type of segmentation task is being evaluated in section 4.2 / Table 2?"
>
> We appreciate the reviewer's query on the discrepancies in Mask2Former's performance metrics. The Mask2Former paper [1] focuses on supervised training across multiple epochs over the entire dataset, whereas our work evaluates **continual learning performance** in segmentation, where classes are introduced sequentially. This sequential approach leads to catastrophic forgetting, significantly reducing accuracies compared to the supervised baseline. For an accurate comparison, please refer to Table 3 in the CoMFormer paper (Cermelli et al., 2023) [2], which benchmarks starting with 100 classes in ADE20K [3] and incrementally adds the remaining 50 classes. We apologize for any confusion and hope this clarifies the matter.
>
> ### **Differences Between Hummingbird and Our Method**
> 3. "The details of how Hummingbird and the proposed method differ have not been discussed making the comparison in Table 5 less useful to the reader. Please elaborate in the manuscript (or appendix)."
>
> Thank you for highlighting the need for a clearer difference between our method and Hummingbird [4] in Table 5. Hummingbird uses nearest neighbor retrieval to predict the classes of patches in target images. In contrast, our method employs kNN to enhance the prediction of each proposed class-agnostic mask. We have updated the manuscript to clarify these differences near Table 5 for better reader understanding.
>
>
> ### **Clarification questions**
> ### **Clarification on Equations and Terms in Section 4.1**
> 1. "In section 4.1, does the "linear combination of original predicted class probabilities, with the retrieved prediction class probabilities" refer to Eq. 4?"
> 2. "In the same paragraph, which equation does the "geometric ensemble" refer to?"
>
> We thank the reviewer for pointing these out! The "linear combination of original predicted class probabilities with the retrieved prediction class probabilities" indeed refers to Eq. 4. We have updated the manuscript to explicitly reference Eq. 4 in that context. Regarding the term "geometric ensemble," originally introduced in FC-CLIP [5], we recognize the confusion and have replaced it with the more intuitive term "late fusion," as commonly referred to in the literature. This change has been made throughout the manuscript.

---

> ### Author Response · Authors · 2024-05-27
>
> ### **Details on Dataset and Annotations in Table 4**
> 3. "In Table 4, which dataset is used to compute the database of embeddings ($D_{vec}$)? Are all methods in the table seeing the same number and types of annotations?"
>
> As stated in Section 4.1, the database of embeddings ($D_{vec}$) is constructed from the training set images and annotations of each evaluation dataset. Consequently, our method differs from others in Table 4 regarding the data seen during training. However, continual learning algorithms often perform worse than zero-shot open vocabulary baselines due to large performance degradation due to catastrophic forgetting – we felt it was important to demonstrate that our method indeed improves performance beyond the best zero-shot baselines.
>
> ### **Broader Impact Statement**
> 4. "Segmentation can be used in sensitive applications such as medical, military, and surveillance. A broader impact statement would be useful."
>
> We appreciate the suggestion to include a broader impact statement. We have added this to the manuscript, emphasizing that since our approach has dual-use applications, we strongly oppose its use in military and surveillance contexts and explicitly prohibit such uses.
>
> ### ***References***
> [1] Cheng, Bowen, et al. "Masked-attention mask transformer for universal image segmentation." Proceedings of the IEEE/CVF conference on computer vision and pattern recognition. 2022.\
> [2] Cermelli, Fabio, Matthieu Cord, and Arthur Douillard. "Comformer: Continual learning in semantic and panoptic segmentation." Proceedings of the IEEE/CVF Conference on Computer Vision and Pattern Recognition. 2023.\
> [3] Zhou, Bolei, et al. "Semantic understanding of scenes through the ade20k dataset." International Journal of Computer Vision 127 (2019): 302-321.\
> [4] Balazevic, Ivana, et al. "Towards in-context scene understanding." Advances in Neural Information Processing Systems 36 (2024).\
> [5] Yu, Qihang, et al. "Convolutions die hard: Open-vocabulary segmentation with single frozen convolutional clip." Advances in Neural Information Processing Systems 36 (2024).

---

> > ### Comment · Reviewer_2MDe · 2024-06-27
> > **Response**
> >
> > Thank you for addressing these concerns and sorry for the delay in my reply.
> >
> > ### Concept customization
> > What's missing is an explanation for what data gather process / dataset, particularly for the computation for $D_{vec}$, and models were used to generate the qualitative results, for example the one for Harry potter and Pokemon. The reader of this manuscript needs to be able to reproduce the results shown in Figure 1.
> >
> > ### Details on dataset and Annotations in Table 4
> > This table then requires an (*) to explain this caveat. I agree that the table is nonetheless useful as it shows how the continual learning regime can be more powerful than the zero shot regime.

---

> ### Author Response · Authors · 2024-07-01
> **Response**
>
> We thank the reviewer for the suggestions.
>
> **Concept Customization**
>
> We agree that the concept customization part needs further clarification, and therefore we changed the caption for Figure 1: "For concept customization, we use EntitySeg (Qi et al., 2022) to generate class-agnostic masks for entities and kNN-CLIP to label these masks. We build the supporting database for each picture by crawling the internet for example images of each class. We then filter the masks from EntitySeg based on confidence thresholds for masks and class predictions."
>
> **Details on dataset and Annotations in Table 4**
>
> We agree with the suggestion and added a footnote for Table 4, which is "Our method aims to continually enhance the performance of open-vocabulary segmentation models. For evaluation, we use the training set of each dataset, which is unseen by other methods, to construct our embedding database. Our approach significantly improves the performance of zero-shot baselines in a continual learning setting."

---

> > ### Comment · Reviewer_2MDe · 2024-07-01
> >
> > Thank you for the quick reply.

---

### Review · Reviewer_Rana · 2024-05-18

**Summary Of Contributions:**

This paper presents a method for open-vocabulary panoptic segmentation, using a combination of CLIP, DINO, and other models. The key idea is to pool the features of labelled objects, to create a database of labelled features, and then use these features to help vote for classification of new query masks. The main benefit here is that new "concepts" can be added by simply appending them to the database, and it is not necessary to fine-tune any model weights.

**Audience:**

Yes

**Claims And Evidence:**

Yes

**Requested Changes:**

I pointed out many weaknesses, but my hunch is that these are not critical weaknesses/errors in method or motivation, but are simply the result of unclear writing. So, these can be addressed by careful rewriting.

Harder to address might be: issues with the stated key contributions, which seem to actually be contributions of prior work.

Finally, it would greatly strengthen the paper if the method were made more clear. What exactly is the input? How many pre-trained models are used, and where do they come from? Once these details are more clear, then it would also be interesting to find out how much specific details matter, like what if different models were used instead? In this sense, it would be great to convey a method that appears more general-purpose.

**Strengths And Weaknesses:**

The paper opens with the claim: "The holy grail in image segmentation is to accurately segment any concept images based on a text query." I don't agree with this, or at least I don't understand it. What is a "concept image" exactly? I don't think related work on image segmentation makes much fuss about concept images.


The key question raised in this work is "Can we enhance these models to a continually-growing vocabulary with data without catastrophic forgetting?" There must be some typo here, or words missing, because I am getting lost in the sentence at the "with data" part. It doesn't seem like the question should be emphasizing this, since the related works all use data as well. The paper makes other surprising claims about data too, like training "to data" (in the next sentence). I cannot make sense of this, except to say that these sentences are broken, and clear meaning is missing.

The authors claim that their approach is "effortless". This does not seem true. The implementation details make it very clear that some effort is required.

The paper claims that its main contribution is to "uncover the paradox where fine-tuning VLM models for segmentation using downstream annotations dramatically reduces their ability to recognize the broad VLM vocabulary". This seems covered by work that was actually described as motivational in this paper, like "CLIP as RNN: Segment Countless Visual Concepts without Training Endeavor" by Sun et al 2023.

The paper claims that its second main contribution is "Training-Free Continual Vocabulary Expansion", but this also is covered by "CLIP as RNN" (which is training-free), and other works like MaskCLIP and ReCO and CaR.


In the method, before describing any database construction, the paper boasts that "One appealing characteristic of our framework is the dynamic extendability of the database and the customization of the database construction based on personalized interest." For a comment like this to make sense, it needs to appear after the introduction of the component being described ("the database").


The paper says that the image dimensions are "usually shrunken by a magnitude of 10 each" when referring to the stride of the ViT. I have actually never seen a ViT with stride 10. What weights are these? Was this ViT trained from scratch?

The paper claims that "efficiently representing images with small feature maps guarantees the memory efficiency of our method". This seems obviously false, because anything can be done afterward to increase memory consumption.

The paper does not make it very clear what exactly the input setup is.  The introduction makes it sound like CLIP is getting turned into an open-vocabulary segmentation model via some labelled dataset, but then  it is revealed that the images already have "query masks", and then the task seems to be merely labelling. Also it's not clear what the model/pre-training setup is: the intro makes it sound like CLIP is the main thing, but then later DINO appears, and some unrelated ViT appears as well, and also a "Mask Proposer" shows up in Figure 3  but never appears in the method.

with it’s corresponding -> with its corresponding


The paper places great emphasis on "we do not store any images for replay", but this is true for all training-free methods. Replay does not make any sense when there's no training.

---

> ### Author Response · Authors · 2024-05-27
>
> Thank you for your thoughtful feedback and constructive suggestions on our manuscript. We appreciate your recognition of the simplicity of our approach. We additionally thank the reviewer for providing detailed feedback to improve clarity. It has improved clarity in areas of the draft! We seek  to address the concerns below:
>
> ### **Introduction Claim**
> - "The paper opens with the claim: "The holy grail in image segmentation is to accurately segment any concept images based on a text query." I don't agree with this, or at least I don't understand it. What is a "concept image" exactly? I don't think related work on image segmentation makes much fuss about concept images."
>
> We agree and have revised the sentence for clarity: "The ultimate goal in image segmentation is to accurately segment any image according to a text query or predefined classes." Thank you for highlighting this.
>
> ### **Key Question on Continual Learning**
> - "The key question raised in this work is "Can we enhance these models to a continually-growing vocabulary with data without catastrophic forgetting?" There must be some typo here, or words missing, because I am getting lost in the sentence at the "with data" part. It doesn't seem like the question should be emphasizing this, since the related works all use data as well. The paper makes other surprising claims about data too, like training "to data" (in the next sentence). I cannot make sense of this, except to say that these sentences are broken, and clear meaning is missing."
>
> We agree and have changed the sentence to “Can we enhance these models to maintain a continually growing vocabulary and prevent catastrophic forgetting?” For the second issue, it is a typo, and the preposition should be “on”. Thank you for pointing this out!
>
> ### **Claim of "Effortless" Approach**
> - "The authors claim that their approach is "effortless". This does not seem true. The implementation details make it very clear that some effort is required."
>
> We agree that the term "effortless" might have been vague and have changed it to "training-free" to accurately describe our approach. Thank you for the suggestion.
>
> ### **First Contribution on Catastrophic Forgetting**
> - "The paper claims that its main contribution is to "uncover the paradox where fine-tuning VLM models for segmentation using downstream annotations dramatically reduces their ability to recognize the broad VLM vocabulary". This seems covered by work that was actually described as motivational in this paper, like "CLIP as RNN: Segment Countless Visual Concepts without Training Endeavor" by Sun et al 2023."
>
> We agree and rephrase our claim! Catastrophic forgetting as a phenomenon is well known broadly [1][2][3][4], and finetuning has been shown to dramatically reduce the open-vocabulary segmentation in Sun et al. [5], however, it was not highlighted. We motivate the necessity for continual learning by showing the surprisingly large degradation of the performance of state-of-the-art model, FC-CLIP, after continual fine-tuning. However, we agree with the reviewer about the novelty of this aspect and removed it from the set of our contributions. We highlight that our primary contribution, as mentioned in the title, is demonstrating effective kNN based training-free continual vocabulary expansion. We argue that it is common for papers to have similar motivation, but it will not degrade our key contribution.

---

> ### Author Response · Authors · 2024-05-27
>
> ### **Training-Free Continual Vocabulary Expansion (Second Main Contribution Question)**
> - "The paper claims that its second main contribution is "Training-Free Continual Vocabulary Expansion", but this also is covered by "CLIP as RNN" (which is training-free), and other works like MaskCLIP and ReCO and CaR."
>
> We highlight the key differences to clarify this misunderstanding. These studies enhance their open-vocabulary performance by utilizing CLIP features more effectively for classification and do not expand the vocabulary, enhancing performance of CLIP segmentation where it is already good. In contrast, we identify classes where CLIP is deficient and enhance performance in these categories; in other words, we intend to expand the vocabularies beyond the vocabulary space of CLIP. Our raw accuracies, hence, are consistently higher than these models as shown in the following table.
>
> |Method | VOC 21 | PC-59 |  A-150 | A-847 | PC-459 |
> |----------|----------|----------|----------|----------|----------|
> | ReCo | 25.1 | 22.3 | 11.2 | - | - |
> | MaskCLIP | 38.8 | 26.4 | 9.8 | - | - |
> | CaR | 67.6 | 39.5 | 30.5 | 8.1 | 13.9 |
> | kNN-CLIP | **85.3** | **62.8** | **41.3** | **17.4** | **19.9** |
>
> The challenge is how to do this? As revised in our introduction, “We observe that continual segmentation methods typically underperform compared to zero-shot open vocabulary models due to catastrophic forgetting. Therefore, we propose adapting the task of continually integrating new classes in continual learning to open-vocabulary segmentation models, offering a more feasible solution for continual segmentation.” Our approach presents a practical alternative in this context.
>
> Motivating continual learning: We state that the vocabulary space of language is **not static**. There are new concepts created every day. For example, the CLIP model trained with data collected in 2021 can never recognize Apple Vision Pro, which is released in 2023. But by simply adding an example of Apple Vision Pro, our method can solve it with no training cost. Meanwhile, in this paper, we identify a potential application, stating, “The dynamic nature of kNN-CLIP makes it suitable for tasks including segmentation in Earth observation in the field of remote sensing (Gui et al., 2024), where a large volume of new data are collected daily. kNN-CLIP is particularly favored in situations where frequent emergence of new data makes retraining models costly.”
>
> ### **Placement of Database Construction Comment**
> - "In the method, before describing any database construction, the paper boasts that "One appealing characteristic of our framework is the dynamic extendability of the database and the customization of the database construction based on personalized interest." For a comment like this to make sense, it needs to appear after the introduction of the component being described ("the database")."
>
> We agree and have moved the comment about the dynamic extensibility and customization of the database to appear after the database component is introduced, as suggested.
>
> ### **Clarification of ViT Stride and Patch Size**
> - "The paper says that the image dimensions are "usually shrunken by a magnitude of 10 each" when referring to the stride of the ViT. I have actually never seen a ViT with stride 10. What weights are these? Was this ViT trained from scratch?"
>
> We agree and have revised the sentence to accurately describe our method: "The embedding dimensions $h'$, $w'$ are determined by the image size divided by the patch size of the vision transformer, which is 14 (the patch size of ViT-G)." We also clarified that our method is completely training-free, using an off-the-shelf FC-CLIP model and DINOv2 ViT-G model. We do not train anything from scratch.

---

> ### Author Response · Authors · 2024-05-27
>
> ### **Memory Efficiency Claim**
> - "The paper claims that "efficiently representing images with small feature maps guarantees the memory efficiency of our method". This seems obviously false, because anything can be done afterward to increase memory consumption."
>
> We agree and have rephrased the statement for clarity: "We reduce space requirements by storing small feature maps instead of input images."
>
> ### **Clarification of Input Setup and Method**
> - "The paper does not make it very clear what exactly the input setup is. The introduction makes it sound like CLIP is getting turned into an open-vocabulary segmentation model via some labelled dataset, but then it is revealed that the images already have "query masks", and then the task seems to be merely labelling. Also it's not clear what the model/pre-training setup is: the intro makes it sound like CLIP is the main thing, but then later DINO appears, and some unrelated ViT appears as well, and also a "Mask Proposer" shows up in Figure 3 but never appears in the method."
>
> We agree and have added a new paragraph at the beginning of the methodology section to clarify the input setup: "We aim to augment existing open-vocabulary segmentation models. These models, including FC-CLIP \citep{fcclip}, which we use as our baseline, usually have a mask proposer trained to generate class-agnostic masks from extracted features. The image CLIP features are pooled using these proposed masks and used to compute similarities with the text CLIP features of class names for classification. Finally, mask predictions and their corresponding class predictions are combined to give the resulting segmentation masks." We have also expanded on our implementation with FC-CLIP in subsection 3.3 and included Fig. 4 for additional clarity.
>
> ### **Typographical Error: "its corresponding"**
> - "with it’s corresponding -> with its corresponding"
>
> We have corrected the typographical error from "it's corresponding" to "its corresponding."
>
> ### **Emphasis on Not Storing Images and Not Replay**
> - "The paper places great emphasis on "we do not store any images for replay", but this is true for all training-free methods. Replay does not make any sense when there's no training."
>
> We clarified the statement to "we do not store any images". Our intent was to emphasize that image storage is not required, and replay is not the focus.
>
> ### **Strengthening the Method Description**
> We agree and have revised the methodology section and subsection 3.3 to provide detailed information on the input, the number of pre-trained models used, and their sources. We believe these updates will enhance the clarity and general-purpose applicability of our method. Thank you for the constructive feedback!
>
> ### ***References***
> [1] Zheng, Zangwei, et al. "Preventing zero-shot transfer degradation in continual learning of vision-language models." Proceedings of the IEEE/CVF International Conference on Computer Vision. 2023.\
> [2] Rebuffi, Sylvestre-Alvise, et al. "icarl: Incremental classifier and representation learning." Proceedings of the IEEE conference on Computer Vision and Pattern Recognition. 2017.\
> [3] Li, Zhizhong, and Derek Hoiem. "Learning without forgetting." IEEE transactions on pattern analysis and machine intelligence 40.12 (2017): 2935-2947.\
> [4] Wu, Yue, et al. "Large scale incremental learning." Proceedings of the IEEE/CVF conference on computer vision and pattern recognition. 2019.\
> [5] Sun, Shuyang, et al. "CLIP as RNN: Segment Countless Visual Concepts without Training Endeavor." arXiv preprint arXiv:2312.07661 (2023).

---

### Decision · Action_Editor_PsQQ · 2024-07-09

**Recommendation:** Accept with minor revision

**Comment:**

After the authors' revision, most of the reviewers' concerns have been addressed. The reviewers unanimously recommend acceptance of the paper.

I find the work valuable and recommend its acceptance and publication.

Additionally, I have the following suggestions for the authors as they prepare the final version:

* Figures 2-4 are not in vector graphics format, causing them to become blurry when the PDF is zoomed in. It is recommended to use vector graphics for these figures.
* A limitations section could be added.
* As suggested by Reviewer Rana, the method could be clarified further.
* Proofread the paper again to avoid any typographical errors.

**Audience:**

Researchers and practitioners working on continual learning, open-vocabulary learning, image segmentation, and vision-language models might be interested in reading this paper.

**Claims And Evidence:**

Summary:

This paper proposes kNN-CLIP, a novel training-free strategy for continual segmentation that is capable of expanding the vocabulary beyond the original vocabulary space of CLIP without catastrophic forgetting. The key idea involves constructing a database of embeddings for representing novel classes and using this database to augment open-vocabulary segmentation models with a KNN-based retrieval mechanism. Evaluations on large-vocabulary semantic and panoptic segmentation tasks demonstrate the effectiveness of the proposed kNN-CLIP.

Claims:

The key claims made in the paper are that the proposed kNN-CLIP strategy is capable of (1) continually expanding the vocabulary of image segmentation models in a training-free manner; (2) preventing catastrophic forgetting; and (3) improving performance on continual semantic and panoptic segmentation tasks across datasets with long-tailed distributions.

Evidence:

The claims are well supported by the characteristics of the proposed method, along with the experimental results presented in both the original submission and the revision.